# Diversity of cortical activity changes beyond depression during Spreading Depolarizations

Azat Nasretdinov[1,7], Daria Vinokurova[1,2,7], Coline L. Lemale [3,7], Gulshat Burkhanova-Zakirova[1], Ksenia Chernova[1], Julia Makarova[4], Oscar Herreras[4], Jens P. Dreier [3,5,6] & Roustem Khazipov [1,2] ✉

Spreading depolarizations (SDs) are classically thought to be associated with spreading depression of cortical activity. Here, we found that SDs in patients with subarachnoid hemorrhage produce variable, ranging from depression to booming, changes in electrocorticographic activity, especially in the delta frequency band. In rats, depression of activity was characteristic of high-potassium-induced full SDs, whereas partial superficial SDs caused either little change or a boom of activity at the cortical vertex, supported by volume conduction of signals from spared delta generators in the deep cortical layers. Partial SDs also caused moderate neuronal depolarization and sustained excitation, organized in gamma oscillations in a narrow sub-SD zone. Thus, our study challenges the concept of homology between spreading depolarization and spreading depression by showing that SDs produce variable, from depression to booming, changes in activity at the cortical surface and in different cortical layers depending on the depth of SD penetration.

Spreading depolarization (SD)[1] is a pathophysiological event critically involved in various brain diseases including ischemic stroke, subarachnoid hemorrhage (SAH), traumatic brain injury (TBI), epilepsy and migraine with aura[2–10]. SD consists of a slowly propagating wave of collective and nearly complete depolarization of neurons and glia, and it is commonly thought to be associated with spreading depression of cortical activity due to a depolarization block, and therefore the terms "spreading depolarization" and "spreading depression" introduced by Leão, their pioneer[1,11], are often used interchangeably[2–8]. Moreover, current clinical algorithms for detecting SD imply a match between the large negative shifts in the extracellular potential characteristic of spreading depolarization and the depression of cortical activity[12]. On the other hand, the SD continuum encompasses a variety of SD propagation patterns in cortical space, notably SDs propagating from superficial layers to different cortical depths[13–19]. However, the question of whether depression is common to all of these SD variants

remains largely unknown. In addition to its conceptual importance, this question is also crucial for both detecting/quantifying SD and stratifying treatment during neurocritical care.

Here, we addressed these issues using full-band ECoG recordings of SDs from patients with SAH, and multisite extracellular/ patch-clamp recordings of high potassium – induced SDs in rats. We provide evidence that SDs produce highly variable, ranging from depression to increase, changes in cortical activity in different cortical layers depending on the depth of SD penetration.

## Results

### Diversity of ECoG activity changes during SDs in human patients with SAH

We first explored changes in cortical activity during SDs using subdural ECoG recordings in sedated patients with SAH from the DISCHARGE-1 cohort[20] (Fig. 1a). SDs were characterized by slow negative DC-voltage

---

[1]Laboratory of Neurobiology, Kazan Federal University, Kazan 420008, Russia. [2]INMED-INSERM, Aix-Marseille University, Marseille 13273, France. [3]Centre for Stroke Research Berlin, Department of Experimental Neurology and Department of Neurology, Universitätsmedizin Berlin, corporate member of Freie Universität Berlin, Humboldt-Universität zu Berlin, and Berlin Institute of Health, D-10117 Berlin, Germany. [4]Department of Translational Neuroscience, Cajal Institute—CSIC, Madrid, Spain. [5]Bernstein Centre for Computational Neuroscience Berlin, D-10115 Berlin, Germany. [6]Einstein Centre for Neurosciences Berlin, D-10117 Berlin, Germany. [7]These authors contributed equally: Azat Nasretdinov, Daria Vinokurova, Coline L. Lemale. ✉e-mail: roustem.khazipov@inserm.fr

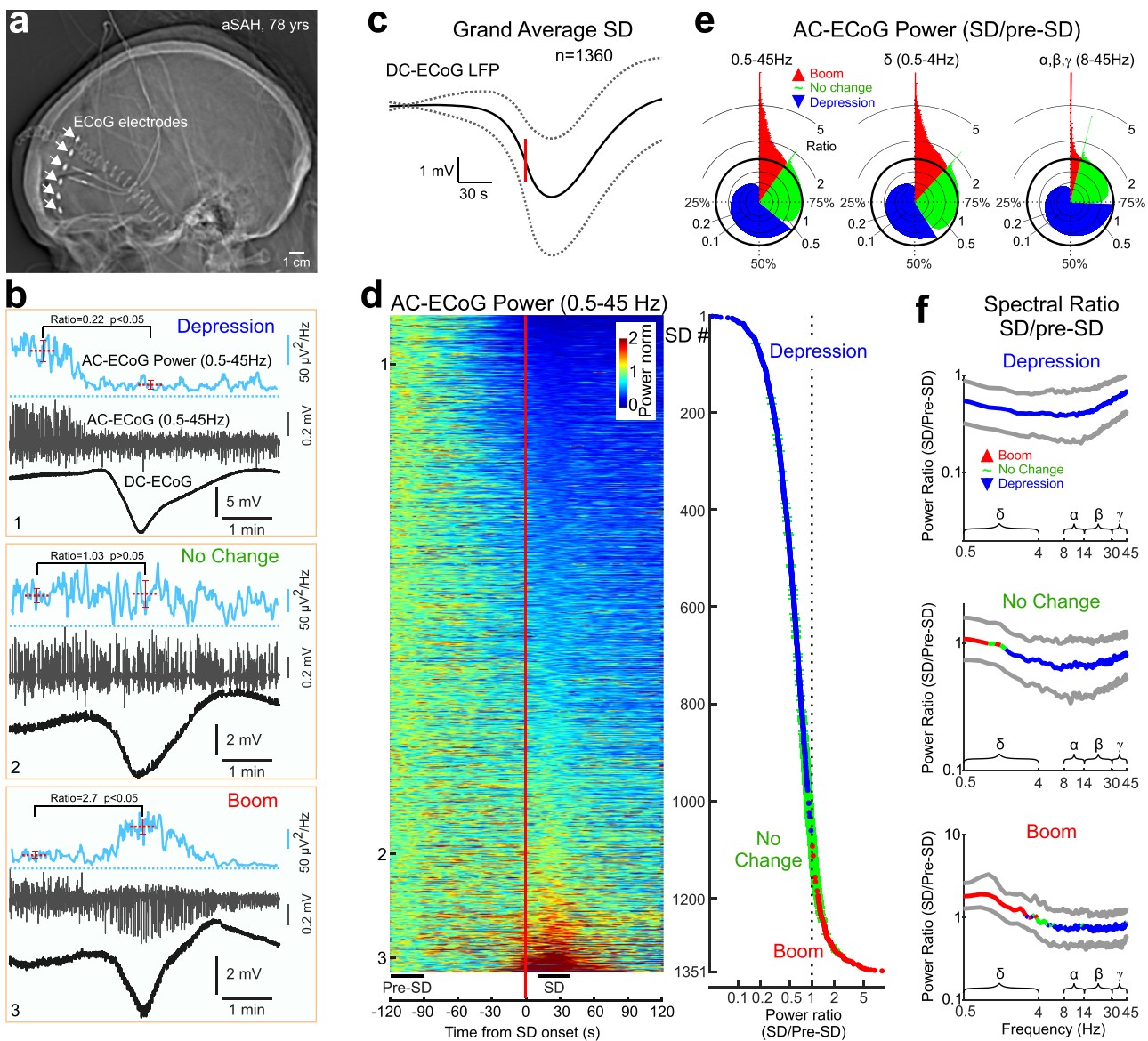

**Fig. 1 | Spreading depolarizations (SDs) exert variable, ranging from depression to boom, effects on electrocorticographic activity in human patients with subarachnoid hemorrhage.** **a** Computer tomography scan showing electrodes positions for subdural electrocorticographic (ECoG) recordings from patients with aneurysmal subarachnoid hemorrhage (SAH). **b** Three SD examples. From top to bottom: AC-ECoG (0.5–45 Hz) power (blue), AC-ECoG (gray) and DC-ECoG (full-band signal; black). Note the variable changes in AC-ECoG activity during SD ranging from depression (top), no change (middle) to boom (bottom). The significance of changes in AC-ECoG power between pre-SD and SD epochs was calculated using the Wilcoxon rank sum two-sided test. **c** Grand average SD with a reference to SD onset (vertical red line), dashed lines show standard deviation. **d** Changes in AC-ECoG power during SD. *Left*, the color-coded AC-ECoG power values are normalized to the pre-SD values, sorted by the level of power changes during SD and plotted with a time reference to SD onset (vertical red line). Each line corresponds to an individual SD. *Right*, corresponding ratios of the AC-ECoG power during SD relative to the pre-SD values. Blue and red dots indicate depression and boom, respectively; green lines indicate no significant change. **e** Pie histograms showing the distribution of the AC-ECoG power during SDs relative to the pre-SD values in the full AC (0.5–45 Hz), $\delta$ (0.5–4 Hz) and fast (8–45 Hz) frequency ranges. **f** Power spectra (median with 25th and 75th percentiles) of the AC-ECoG activity during SD normalized to the pre-SD values in groups showing depression (top), no change (middle) and boom (bottom) in the full AC frequency range (Wilcoxon signed-rank two-sided test). **d**, **f** Pooled data from $n = 1351$ SDs recorded from 26 aSAH patients. Source data are provided as a Source Data file.

shifts (Fig. 1b, c), and variable changes in activity in the conventional EEG frequency bands (0.5–45 Hz, AC-ECoG). Besides "classic" cases of SD associated with depression of AC-ECoG activity (Fig. 1b, example #1 & Supplementary Fig. 1a), we also observed SDs without significant changes in AC-ECoG activity power (Fig. 1b, example #2 & Supplementary Fig. 1b), and cases of SD associated with a paradoxical transient increase in cortical activity, hereafter referred to as a "boom" (Fig. 1b, example #3 & Supplementary Fig. 1c). We quantified the occurrence of these different SD phenotypes by sorting SDs by the change in AC-ECoG activity power during SD relative to baseline pre-SD values as shown in the raster plot of all SDs and considering the

significance of these changes (Fig. 1d). SDs with varying degrees of depressed AC-ECoG activity accounted for 64% of SDs, with 26% of SDs showing no significant change, and 10% of SDs showing a boom in activity (Fig. 1e). Changes in the slow $\delta$-frequency band accounted for most of the variability in changes in AC-ECoG activity during SDs, while fast activities in $\alpha-$, $\beta-$ and $\gamma-$ frequency bands showed a more uniform depression in all types of SD (Fig. 1f). Correspondingly, sorting SDs along the change in $\delta$- frequency power revealed a greater frequency of "no change" and "boom" SDs than in the full AC-ECoG band, while sorting along the change in fast activity power revealed a greater prevalence of SDs with depression (Fig. 1e). Thus, changes in AC-ECoG

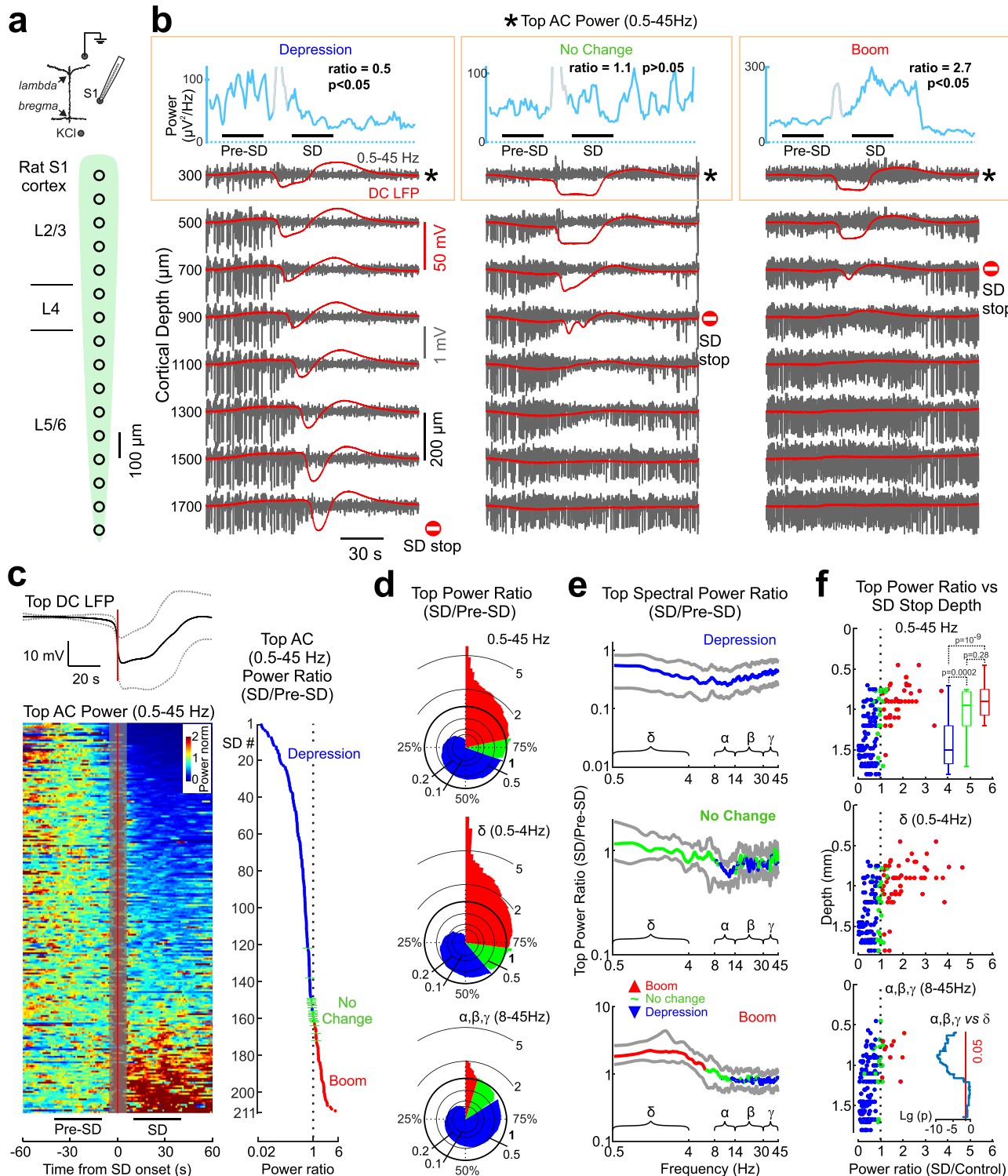

activity during SD in patients with SAH were not limited to depression, but ranged from different levels of depression to boom, and were more specific to the δ-frequency band.

## Diverse changes in electrical activity at the rat cortical vertex during SDs

Next, we wondered whether the variability in changes in electrical activity during SDs is also present in the cortex with fully normal metabolism. To this end, we investigated SDs induced by distant high-potassium application and recorded using silicon probes in the urethane-anesthetized rat somatosensory barrel cortex (Fig. 2a). As in human patients, SDs were associated with various changes in the field potential (FP) activity on the upper electrodes located close to the cortical surface (Fig. 2b). These included cases of SD with profound depression of activity in the AC (0.5–45 Hz) band (Fig. 2b, left), cases with minor changes (Fig. 2b, middle), and cases with increased activity during SD (Fig. 2b, right). Depression was observed more frequently (70% of SDs), but the level of depression was highly variable; and in a significant number of SDs there was either no change (8%) or an increase (22%) in FP activity power in the AC-band (Fig. 2c, d). SD-related changes in cortical activity were more specific to the δ-frequency band (Fig. 2e), and the incidence of SD cases with no change

**Fig. 2 | Changes in electrical activity at the cortical vertex during high potassium-induced SD depend on the depth of SD propagation. a** Scheme of recordings from different depths of the rat barrel cortex. SDs are evoked by distant epipial high-KCl application. **b** Three SD examples. The top blue trace shows the AC (0.5–45 Hz) LFP power at the top electrode. The SD onset is shaded to discard spectral leakage from SD rise front. Red traces: DC-LFP (full-band), gray traces: AC-LFP (0.5–45 Hz) at different cortical depths. Stop sign: SD stop depth. **c** Changes in AC-LFP power at the top electrode during SD. *Top*, grand average SD with a reference to SD onset (vertical red line), dashed lines show standard deviation. Left, color-coded AC-top LFP power values are normalized to the pre-SD values, sorted by the level of power changes during SD and referenced to SD onset. Each line corresponds to an individual SD. *Right*, corresponding SD/pre-SD ratios of the AC-top LFP power. Blue, red and green dots indicate depression, boom and no significant change, respectively. **d** Pie histograms of the AC-top LFP power SD/pre-SD ratios in different frequency ranges. **e** Power spectra (median, Q1/Q3) of the AC-top LFP during SD normalized to the pre-SD values in groups showing depression (top), no change (middle) and boom (bottom) in the full AC frequency range (Wilcoxon signed-rank two-sided test). **f** Dependence of the SD/pre-SD ratio in the AC-top LFP power on the SD stop depth in the full AC (top), $\delta$ (0.5–4 Hz) and fast (8–45 Hz) frequency ranges. Each dot corresponds to an individual SD. The inset in the top plot shows boxplots [center line, median; edges, Q1/Q3; whiskers, non-outlier extremes] of the SD stop depth distribution for SDs with AC-top LFP depression (blue), no change (green) and boom (red); Kruskal-Wallis test. The inset in the bottom plot shows the depth-dependence of $p$ values for a difference between AC-top LFP power in the $\delta$ (0.5–4 Hz) and fast (8–45 Hz) frequency ranges (Wilcoxon rank sum two-sided test). **c–f** Pooled data from $n = 211$ SDs from 13 rats. Source data are provided as a Source Data file.

and booming was higher in the $\delta$- frequency than in the full AC-band, whereas fast activities showed a higher incidence of depression (Fig. 2d). Overall, these results showed a high degree of similarity between SDs in human patients and high potassium-induced SDs in the rat, including a wide range of SD-related changes in activity at the cortical surface involving various grades of depression, no change and booming, and relative specificity of these changes to the $\delta$- frequency band.

## Dependence on the depth of SD propagation
Previous studies have shown a diversity of SD propagation profiles, with a prevalence for SD onset in superficial layers, a top-down SD propagation gradient and SD stopping at different cortical depths[13–19,21]. We found a strong correlation between the vertical SD profile and changes in cortical activity at the cortical surface. Indeed, full SDs propagating throughout the cortex were associated with deep depression, whereas SDs partially penetrating the cortex were associated with little change or an increase in FP power at the surface electrodes (Fig. 2b). Along with these observations, the ratio of cortical surface activity power in the AC-band during SD to baseline activity before SD was negatively correlated with the depth of SD spread (R = −0.59, $p < 0.001$; $n = 211$ SDs; Fig. 2f, top panel). This relationship was steeper for the power in the $\delta$- frequency band, and shallower for the power of fast (8–45 Hz) activity (Fig. 2f, middle panel). In addition, SD cases with depression, no change and boom of activity at the cortical surface showed a different distribution of SD stop depth values from bottom to top in cortical depth (Fig. 2f, bottom panel). Thus, the variability in vertical SD propagation profiles largely accounted for changes in AC-activity at the cortical surface during high-potassium-triggered SDs, which switched polarity from boom to depression with increasing SD penetration depth, and most specifically involved activity in the $\delta$− frequency band.

## Source of persistent $\delta$-activity during partial SDs
Electrical potentials recorded at the cortical surface result from complex summation of multiple current generators located in different cortical layers[22,23]. Current source density (CSD) analysis of the UP-states of $\delta$−oscillations in the rat cortical barrel column revealed complex source/sink profiles spanning the supra- and infragranular layers under control conditions (Fig. 3a, b). $\delta$−oscillations were completely suppressed through all cortical depth during full SDs propagating throughout the cortex (Fig. 3a). During partial SDs, $\delta$−oscillations persisted but their CSD was strongly simplified: only a deep sink/source dipole remained, and it was associated with large positive delta waves from mid-layers up to the surface (Fig. 3a, b). Since CSD profiles in the cortex display multiple overlapped poles arising from activity elicited by different pathways along the cortical width and the respective dipoles occlude partially to each other[24] we carried out a decomposition of FP profiles into spatially-coherent FP generators involved in $\delta$−oscillations using independent component

analysis (ICA)[25]. Four principal FP generators jointly accounted for >99% of total variance during control activity. Three presented a characteristic voltage profile (Fig. 3c, d) with maxima within the cortical width (cortical generators) and another exhibited a flat profile and thus was considered a generator originating remote (R) from the recording array. The strongest was designated main (M) because it accounted for >60% of the total variance, had a smooth wide maximum spanning layers 4-5 and a polarity reversal at 300–400 μm under the pia. The two other cortical generators were labeled as L3 and L5/6 according to the layers displaying the maxima. These FP generators were differently affected during SDs propagating to different cortical depths (Fig. 3c, e, f). During full SDs all generators except the R-generator were suppressed, and the latter was also unaffected during partial SD, as expected since it originates outside the cortex. Yet, partial SDs suppressed all cortical generators except L5/6 (Fig. 3c, e, f), whose second spatial derivative (akin to CSD) presents the same profile than the group of currents spared during partial SDs. Moreover, L5/6 generator showed, on average, nearly two-fold increase of the power during superficial SDs (Fig. 3f). This identifies L5/6 as the FP generator whose passive currents oriented towards the pia spread $\delta$-activity upwards through the volume conduction so accounting for the SD-spared $\delta$-activity from middle layers up to the surface during partial SDs. To verify this hypothesis, we inhibited deep cortical layers by local injection of the GABA(A) receptor agonist isoguvacine. Isoguvacine microinjection at depth -1.3 mm induced long-lasting MUA suppression in deep cortical layers (to <10% of control values) without significant change in MUA levels in the superficial layers (Supplementary Fig. 2a, b), and suppressed $\delta$-FP power and $\delta$-generators through all layers (Supplementary Fig. 2c, d). Isoguvacine also strongly reduced the changes in $\delta$-activity both during full and superficial SDs, and eliminated an increase in $\delta$-activity at the top electrodes during superficial SDs (Supplementary Fig. 2e, f) supporting the hypothesis that persisting $\delta$-activity at the cortical surface during partial SDs in control conditions is supported by spared activity in deep layers.

## Neuronal activity during SDs with different vertical propagation profiles
We further explored changes in neuronal activity across cortical layers during SDs of different vertical propagation profiles. In all cases, SD was preceded by a brief phase of increase in multiple unit activity (MUA, Pre-SD excitation[26–28],) followed by complete suppression of MUA in the layers invaded by SD (Fig. 4a). This wave of Pre-SD excitation propagated vertically along with SD and organized into local fast $\gamma$-oscillations[26,28–30] (Fig. 4a, b). During partial SDs, the cortex within -300 μm just below the SD stopping depth, showed persistently increased firing throughout the SD in the cortex above (Sub-SD excitation zone Fig. 4a, c)[14], that was also organized into local fast $\gamma$-oscillations along with suppression of $\delta$−activity (Fig. 4b–d). The cortex below this Sub-SD excitation zone exhibited an unchanged MUA associated with ongoing $\delta$−activity (Fig. 4a, c, d). During full SDs, the

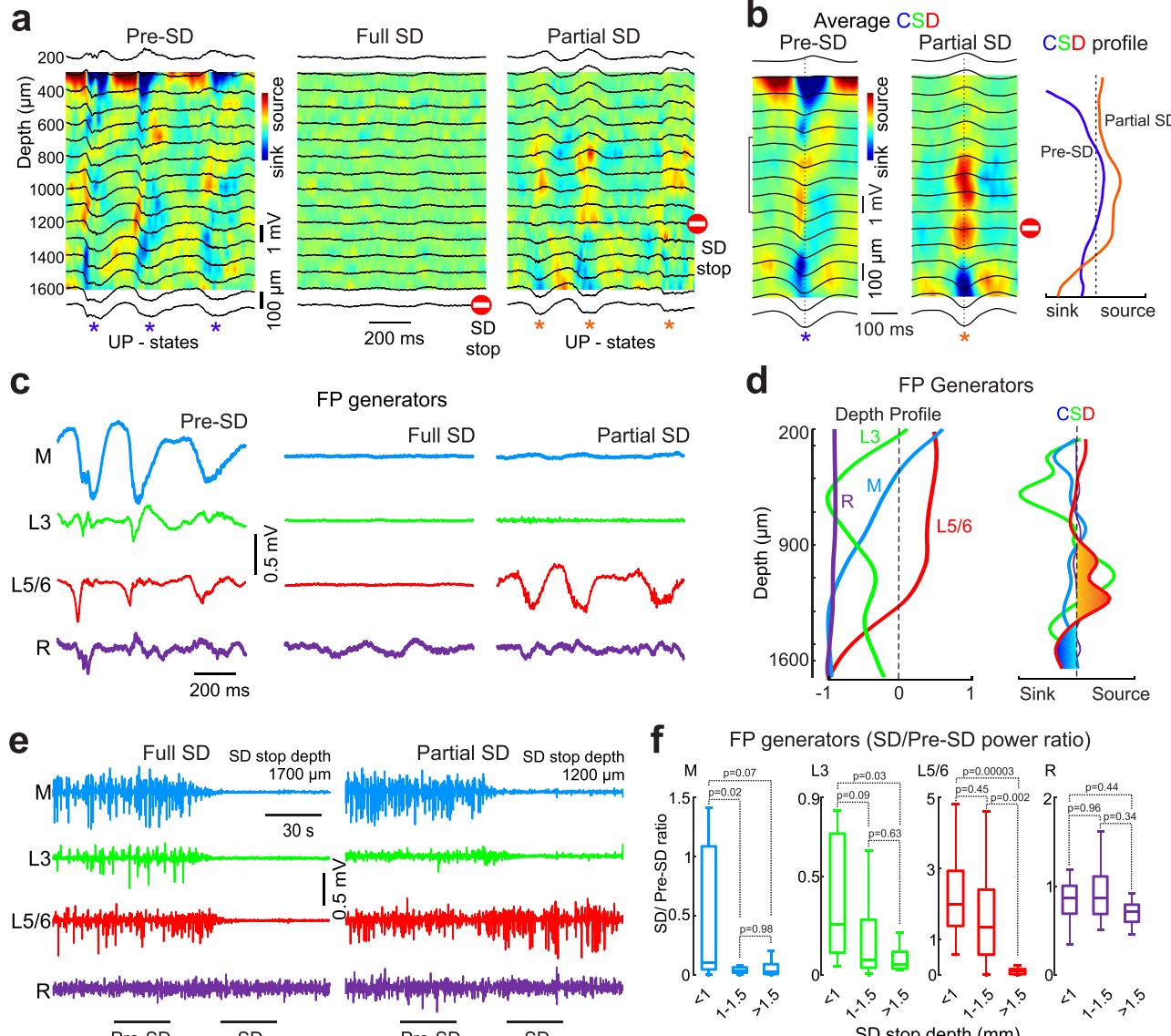

**Fig. 3 | Current source density profile and field potential generators of δ-oscillations during SDs. a** Example traces of FP (black) at different cortical depths overlaid on current-source density (CSD) map in pre-SD epoch (left), during full SD (middle) and partial SD terminated at depth of 1200 µm (right). **b** Average UP-states of δ-oscillations during pre-SD epoch and partial SD (left), and corresponding CSD values at the peak of the UP-states (right). **c** Example traces of the four main FP generators obtained by the independent component analysis (ICA) of FP profiles. **d** Left, Characteristic spatial profile (left) of each component of the FP generator (Main (blue), L3 (green), L5/6 (red) and Remote (purple)). Right, the second

derivative of the spatial profiles (right) as an analog of CSD. **e** The time course of each of the generators during a full SD (left) and partial SD (right) stopped at a depth of 1200µm. **f** Grouped data from 11 rats ($n = 180$ SDs) presented as boxplots [center line, median; edges, Q1/Q3; whiskers, non-outlier extremes] for the SD/pre-SD power ratio (values < 1 correspond to a depression, >1 refers to a boom) for each of the generators at three ranges of SD stop depth: less than 1 mm (surface Partial SDs), between 1 and 1.5 mm (intermediate Partial SDs), and deeper than 1.5 mm (Full SDs), Kruskal-Wallis test. Source data are provided as a Source Data file.

wave of Pre-SD γ-excitation propagated vertically along with SD throughout entire cortical depth followed by complete suppression of MUA in all layers.

Consistent with these findings, whole-cell recordings of L5 neurons, identified as regular spiking (RS, $n = 5$), intrinsic bursting (IB, $n = 4$) and fast-spiking (FS, $n = 1$) neurons (Supplementary Fig. 3 and Supplementary Table 1) revealed variable changes in membrane potential and action potential (AP) firing depending on the SD stop depth (Fig. 5a–d and Fig. 6, Supplementary Table 2). Full SDs and SDs that propagated into the cortex deeper than the site of the cell being recorded were associated with profound depolarization to −26 ± 15 mV (by 34 ± 13 mV from the pre-SD values, $n = 26$ SDs) and AP inactivation preceded by a transient excitation phase at the beginning of the SD, when neurons emitted short bursts of APs (Fig. 5a, b) On average, AP

inactivation occurred at a membrane potential of −36 ± 6 mV ($n = 26$ SDs) (Fig. 5e), consistent with previous reports[27]. However, during partial SDs stopping in their vertical propagation just above the neuron being recorded, membrane potential exhibited only moderate depolarization and neurons displayed sustained AP firing throughout the duration of the SD above (Fig. 5c, g) with maximal depolarization to −47 ± 6 mV (by 18 ± 4 mV from the pre-SD values, $n = 11$ SDs; Fig. 5e). SDs stopping far above the recorded neuron were associated with little depolarization through the SD above and small reduction in the UP-states amplitude (Figs. 5d–f, 6). Consistent with these observations, the level of neuronal depolarization and the duration of AP firing during SD strongly depended on the vertical distance between the cell and the SD stop depth (Fig. 5f, g). Thus, different subtypes of SDs were associated with complex changes in membrane potential, AP firing and

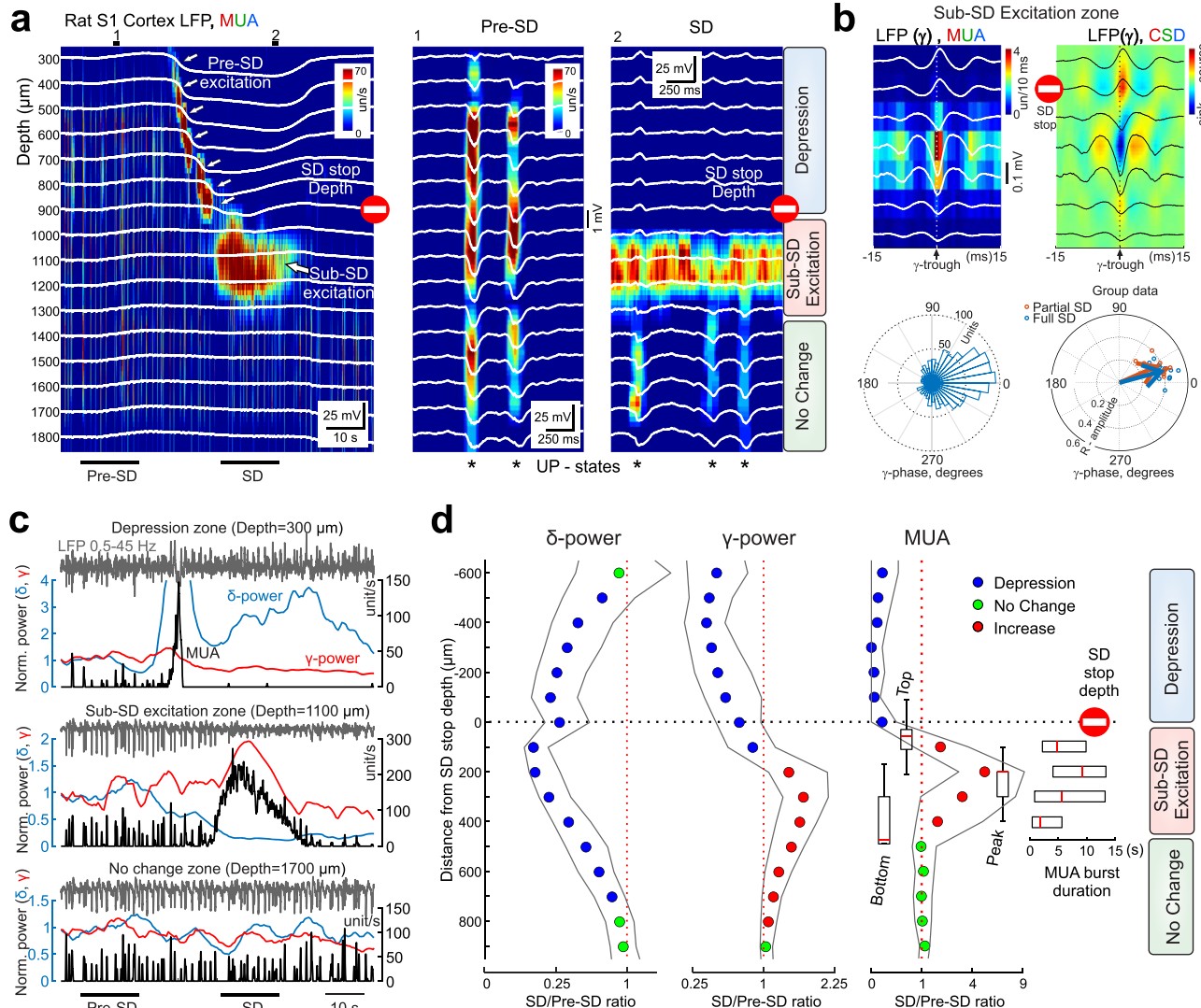

**Fig. 4 | Three vertical zones formed in a cortical column during partial SDs. a** Partial SD stopping at 900 μm from the cortical surface. DC-LFPs at different cortical depths (white traces) are overlaid on color-coded multiple unit activity (MUA) density map. Example recordings before (1) and during SD (2) at the expanded time scale. *: UP-states of δ−oscillations. Note three zones formed during SD: upper zone with depressed MUA, intermediate sub-SD zone with sustained neuronal excitation and deep zone which maintains δ−oscillations and MUA. **b** γ-trough triggered average depth profiles of LFP, MUA density and CSD in the sub-SD excitation zone during partial SD. Below, corresponding circular histogram of LFP phase preference for L5 MUA in Sub-SD excitation zone during partial SD (left) and group data for L5 MUA phase preference for gamma oscillations during pre-SD excitation during full SDs (blue circles) and Sub-SD excitation during partial SDs (red circles); arrows show mean vectors (*n* = 11 animals). **c** Corresponding to the panel a: LFP (0.5–45 Hz, gray), LFP δ−power (blue) and γ−power (red) normalized to pre-SD values and MUA (black) in the three zones formed during partial SD: upper zone of depressed MUA (top), sub-SD excitation zone (middle) and under-SD zone of unchanged MUA (bottom). **d** Dependence of the change in δ−power (left) and γ−power (middle) and MUA frequency (right) during SD through the cortical depth relative to the depth of SD stop. Note the depression (blue, *p* < 0.05, Wilcoxon signed-rank two-sided test) of δ− and γ− activity and MUA in the upper zone; depression of δ− but increase in γ− activity and MUA (red, *p* < 0.05) in the sub-SD excitation zone; and unchanged activity (green, *p* > 0.05) in the deep zone. Insets: vertical boxplots of group data [center line, median; edges, Q1/Q3; whiskers, non-outlier extremes] for depth of top and bottom borders, and peak MUA increase in the sub-SD excitation zone; horizontal boxplots [center line, median; edges, Q1/Q3] for duration of excitation in the sub-SD zone. Pooled data from 102 SDs terminating at depth from 450 to 1250 μm. Source data are provided as a Source Data file.

neuronal network activity in different cortical layers depending on the vertical profile of SD propagation. While full SDs were associated with profound depolarization and AP inactivation due to the depolarization block, partial SDs were associated with variable levels of neuronal depolarization that may or may not reach depolarization block depending on the vertical distance of the neuron from the depth of the SD stop.

## Discussion
The main findings of the present study are as follows: (i) changes in AC-ECoG activity during SD in patients with SAH were not limited to

depression, but ranged from different levels of depression to boom, and were more specific to the δ-frequency band; a similarly wide range of SD-related changes in activity at the cortical surface and relative specificity of these changes to the δ-frequency band was also characteristic of high potassium-induced SDs in the rat; (ii) variability in vertical SD propagation profiles largely accounted for changes in AC-activity at the cortical surface, which switched polarity from boom to depression with increasing SD penetration depth. Depression of activity at the cortical surface was characteristic of full SDs penetrating all cortical layers. In contrast, partial SDs, confined to the superficial layers, were associated with no changes or boom of activity at the

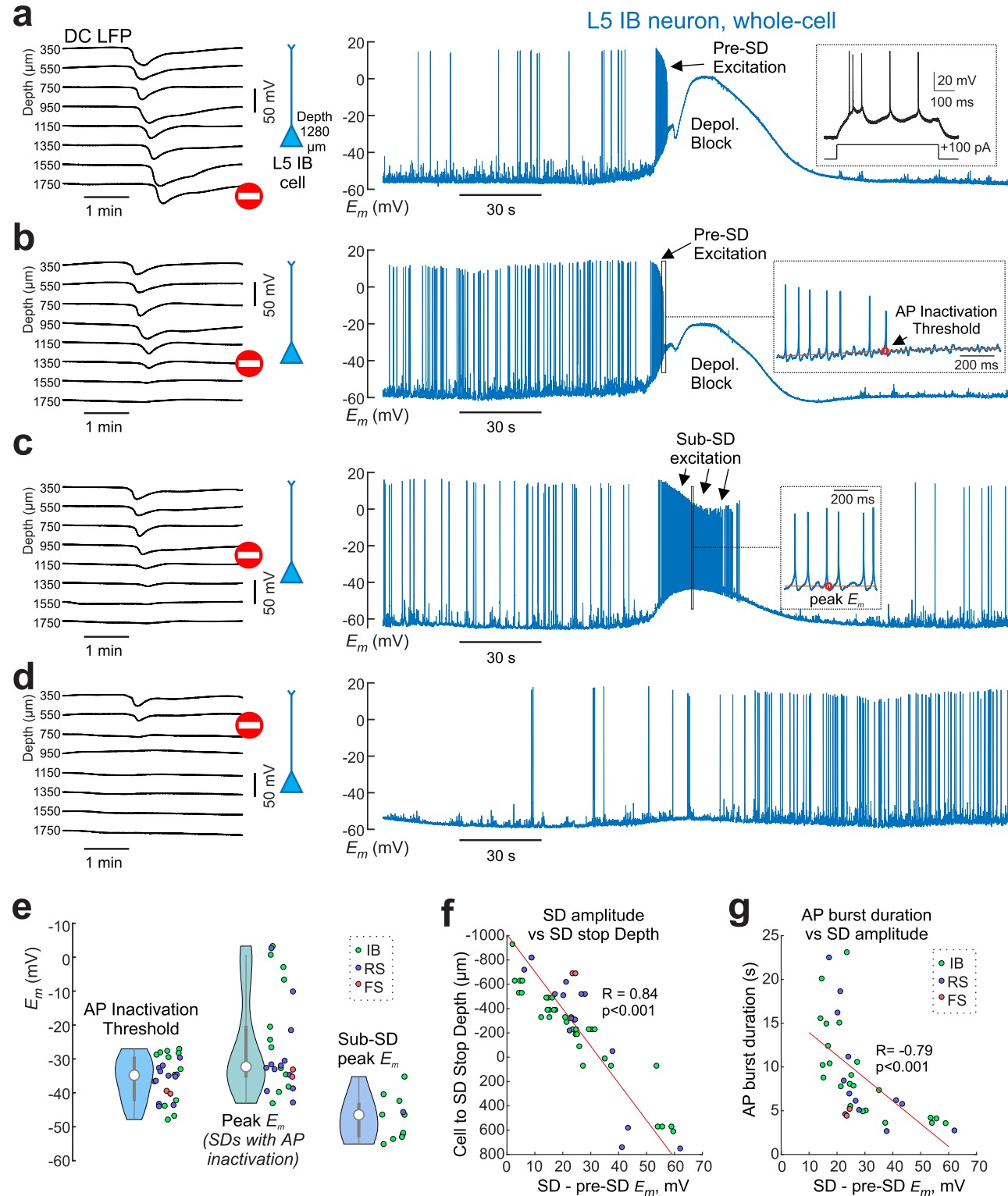

cortical surface, supported by volume conduction of signals from spared δ- oscillators in the deep layers. (iii) at cellular level, full SDs were associated with profound depolarization and AP inactivation due to depolarization block, whereas partial SDs were associated with variable levels of neuronal depolarization depending on the vertical distance of the neuron from the depth of the SD stop. In a narrow sub-SD zone, partial SDs caused mild depolarization and sustained excitation of neurons, organized in network gamma oscillations. Taken together, these findings challenge the concept of homology between spreading depolarization and spreading depression by showing that

SDs produce variable, from depression to booming, changes in activity at the cortical surface and in different cortical layers depending on the depth of SD penetration.

## Revisiting homology between spreading depression and depolarization

Historically, spreading depression of cortical activity was first described by Leão using AC-ECoG recordings[11]. Later, Leão demonstrated that the spreading depression of cortical activity is associated with the SD wave characterized by a negative DC-shift of extracellular potential

**Fig. 5 | Variable dynamics of membrane potential and firing in deep neurons during SDs with different propagation profiles.** Examples of four SDs stopping at different cortical depths (left. DC-LFP traces) and concomitant whole-cell recordings (right) of the membrane potential in a L5 IB neuron (cell #3 in Supplementary Table 1). SDs are selected from a cluster of SDs and sorted by the depth of SD penetration; (**a**), SD #13; (**b**), SD #4; (**c**), SD #1; (**d**), SD #3 (see also Supplementary Table 2). Insets: (**a**) response to depolarizing current step during pre-SD epoch; (**b**), example recordings of the membrane potential at the beginning of SD when the cell falls into depolarization block; (**c**) example recordings of the membrane potential at the peak depolarization in the case of an SD stopping -200 µm above the cell. **e** Violin plots [center circle, median; thick gray whiskers, Q1/Q3; thin gray whiskers,

non-outlier extremes] of the AP depolarization block threshold values ($n = 26$ SDs), peak membrane potential values during SDs with AP depolarization block ($n = 26$ SDs), and peak membrane potential values during partial SDs terminating above L5 ($n = 11$ SDs). **f** Dependence of the membrane potential depolarization during SD from pre-SD values on a vertical distance between the recorded cell and the depth of SD stop. Pooled data from 52 SDs ($n = 10$ cells from 9 rats; IB, green; RS, blue; FS, red; R, Spearman's correlation coefficient; $p = 10^{-14}$). **g** Dependence of the AP burst duration on a depolarization in L5 neurons during SD. Pooled data including pre-SD and sub-SD excitation bursts from 38 SDs ($n = 8$ cells from 7 rats; R, Spearman's correlation coefficient; $p = 10^{-7}$). Source data are provided as a Source Data file.

during full band recordings, and suggested that spreading depression is caused by a profound depolarization and inactivation block of cortical neurons during SD[1]. These observations formed the basis of the textbook concept of homology of these two phenomena, manifested in the interchangeable use of the terms "spreading depolarization" and "spreading depression", and exacerbated by the frequent use of the same acronym (SD, or cortical SD (CSD))[2–8]. While our results are consistent with Leão's idea that spreading depression is caused by SD, we found that not every SD causes spreading depression. Instead, SD appears to be a more diverse phenomenon, not limited to depression, but including a rich repertoire of activity changes ranging from depression to boom, and varying in different cortical layers depending on the depth of SD propagation.

**Generative mechanisms of cortical activity during SDs**

Slow ($\delta$−band) cortical activity characteristic of sleep and other states of reduced consciousness, which contributed most to the changes in AC-ECoG activity during SDs, was predominant in our recordings from sedated patients and anaesthetized animals. $\delta$−oscillations are internally generated in cortical and thalamocortical networks and support large-scale horizontal synchronization of cortical activity[31–38]. They are supported by collective fluctuations of cortical neurons between UP and DOWN states driven by intracortical excitatory and inhibitory synaptic connections and thalamocortical inputs[34,38–45]. Consistent with previous studies, background $\delta$−activity during pre-SD epochs resulted from a summation of multiple $\delta$−generators, each of which was characterized by a particular CSD and FP profile[25,43,46–50]. During SDs, these generators were differently affected depending on the depth of SD propagation. During full SDs, all active $\delta$−generators were suppressed along with silencing through the entire cortical depth. However, during partial SDs, only superficial $\delta$−generators were selectively suppressed whereas deep $\delta$−generators persisted, and their volume conducted FP signals primarily contributed to the activity in superficial layers and cortical surface. Spared during partial SDs, the deep $\delta$-generator likely relies on minimally altered activity in the deep sub-SD zone, supported by spared local circuitry and thalamocortical inputs. Accordingly, suppression of deep cortical levels with iso-guvacine eliminated an increase in $\delta$− power at the cortical surface during partial SDs. These observations are consistent with a persistence of $\delta$−oscillations in the superficial layers produced by volume-conducted signals from deep $\delta$−generators previously observed after pharmacological suppression of superficial $\delta$−generators by epipial application of the sodium channel blocker lidocaine and glutamate receptor antagonists[25,51]. A similar paradigm has also been described in relation to epileptiform activities, where partial SDs are associated with robust epileptiform signals in the superficial layers and at the cortical surface, resulting from the volume propagation of the epileptic discharges generated in the deep layers spared from SD[19]. In keeping with stronger impairments in the superficial layers following focal ischemia[21], in future studies, it would be interesting to verify whether a similar mechanism explains the "slowing down" of EEG in the ischemic cortex characterized by an increase in $\delta$−band activity concomitant with a decrease in the power of fast activities (reduction of

$\alpha/\delta$−ratio)[52,53]. A paradoxical boom in $\delta$−band activity at the cortical surface, which was characteristic of partial SDs with minimal SD penetration, remains only partially understood, however. The parsimonious explanation could involve the suppression of superficial, electronegative at cortical surface dipoles during partial SDs, which partially mask surface-positive deep UP-state dipoles[25,54]. Furthermore, superficial SDs have been associated with an enhancement of deep L5/6 $\delta$−generator. However, FP $\delta$−power and neuronal depolarization during UP-states in deep neurons slightly decreased during superficial SDs. Therefore, an apparent increase in L5/6 cortical $\delta$−generator could be due to suppression of the superficial component of the Main generator and leakage of its deep component to the L5/6 generator. Alternatively, suppression during partial SD of complex multipolar currents associated with the Main generator would unmask the actual (larger) magnitude of currents associated with the spared L5/6 cortical $\delta$-generator, supporting both the apparent increase of this generator and the paradoxical boom of $\delta$−activity during superficial SDs[25,51]. In future research, it would be interesting to verify the mechanisms underlying SD stop depth-specific changes in cortical activity across cortical layers, including the activity boom at the cortical surface during partial SDs using a computational modeling approach[10,25,55–58].

In addition to these complex changes in the $\delta$−frequency band, SDs were also associated with local (200-300 µm) neuronal excitation organized in $\gamma$−oscillations. These were observed in a form of brief MUA bursts organized in $\gamma$−oscillations at the SD onset in the layers invaded by SD[26–30], and sustained elevation of MUA patterned by $\gamma$−oscillations in the Sub-SD zone through the entire time course of SD in the layers above the SD stop depth[14]. Both types of $\gamma$−oscillations during SDs were associated with neuronal depolarization and firing with a difference that brief $\gamma$−oscillations at the SD onset were curtailed by further neuronal depolarization and AP inactivation, whereas sustained $\gamma$−oscillations in the sub-SD zone were associated with sustained moderate depolarization which did not reach the depolarization block levels. Mechanisms of $\gamma$−oscillations at the SDs' onset and in the sub-SD zone likely involve mutual excitatory/inhibitory synaptic interactions, gap junctions and intrinsic subthreshold membrane potential oscillations[29,30,59]. Of note, because of the limited extent of $\gamma$−oscillations, their volume-conducted signals poorly propagate at a distance and are therefore barely visible at the cortical surface[25,60].

The question arises, why SD phenotypes with unchanged or increased activity at the cortical surface have not been reported before? Because the changes in AC-ECoG activity critically depend on the depth of SD penetration, we suggest that previous SD descriptions mainly dealt with full SDs and the depression phenotype, whereas the dataset used in the present study involved SDs organized in clusters. In SD clusters, the first SD is typically full, whereas the occurrence of partial SDs increases with advancement within the SD cluster[17,18]. Second, in keeping with initial Leão's observations[1,11], SDs may compartmentalize in superficial layers upon their horizontal propagation due to higher propensity of superficial layers to SD[17,61–64], and a "barrier" in vertical top-down SD propagation at around a border between superficial and deep layers[15,18]. Taking into account a relationship

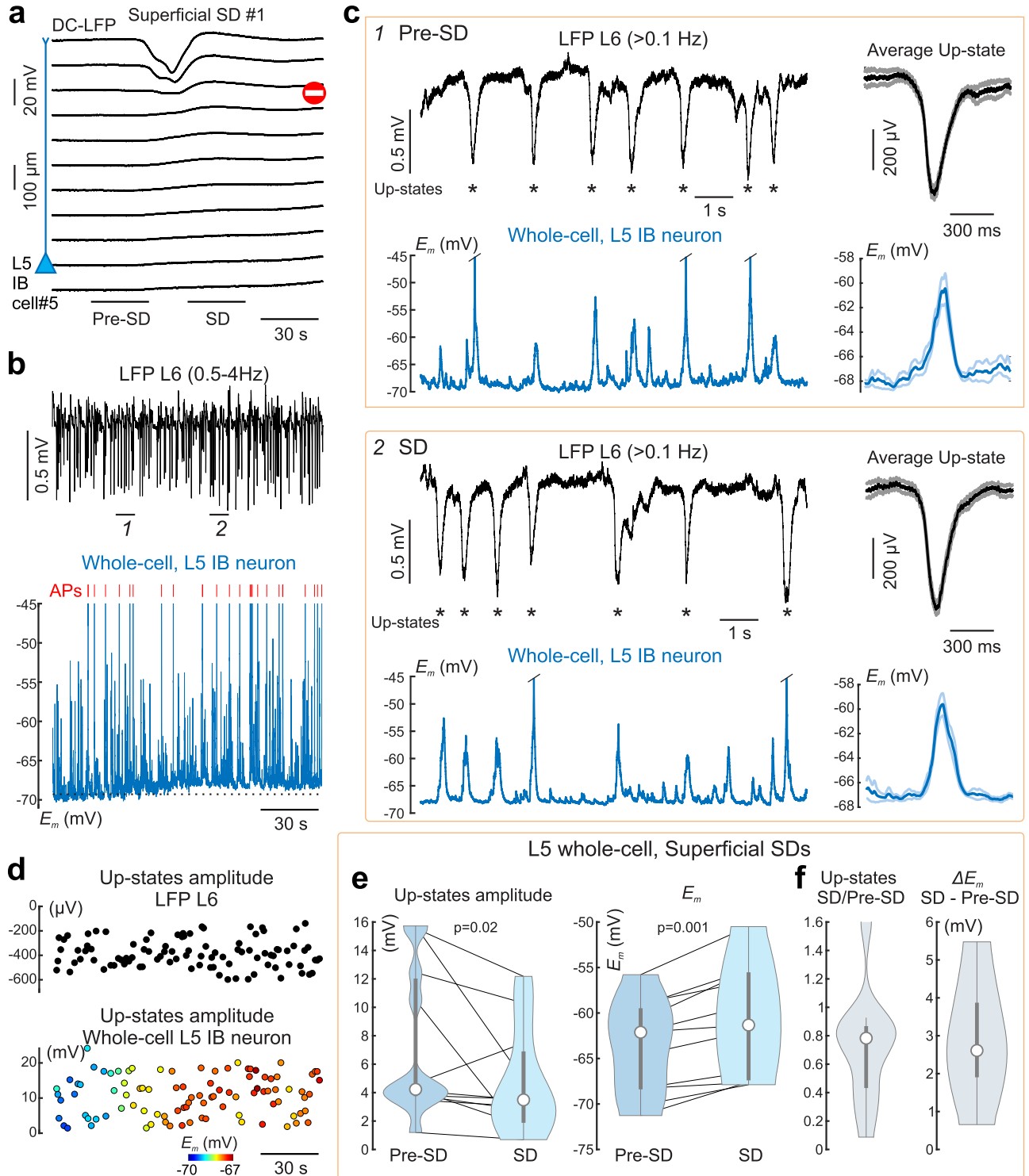

**Fig. 6 | Changes in membrane potential and UP-states in deep neurons during superficial SDs.** **a** Example DC-LFP recordings of a partial SD (#1) confined to the superficial layers. Left, position of the patch-clamped L5 neuron (IB, cell#5). **b** Corresponding LFP recordings (0.5–4 Hz) from L6 (top black trace) and whole-cell current-clamp recordings from the L5 neuron (blue trace) during partial SD presented in (**a**). Red bars indicate APs truncated for clarity. **c** Example traces of L6 LFP and $E_m$ of L5 neuron before (episode #1) and during SD (episode #2) on an expanded time scale, as indicated in (**b**). Asterisks indicate UP-states detected from LFP recordings. Right, corresponding UP-state – triggered averages of L6 LFP and $E_m$ before ($n = 20$ UP-states) and during SD ($n = 20$ UP-states). **d** Amplitude of UP- states L6 LFP and L5 $E_m$ (time interval corresponds to (**a**, **b**)); color code for $E_m$ values indicates resting membrane potential values. **e** Violin plots of group data [center circle, median; thick gray whiskers, Q1/Q3; thin gray whiskers, non-outlier extremes] for L5 $E_m$ UP-state amplitude (*left*) and resting membrane potential (*right*) before and during SD (Wilcoxon signed-rank two-sided test, $n = 11$). **f** Left, violin plot of SD/pre-SD ratio for L5 $E_m$ UP-state amplitudes. Right, violin plot showing depolarization of the resting membrane potential in L5 neurons during SD. **e**, **f** Pooled data from three L5 neurons and 11 SDs which stopped $700 \pm 300\,\mu m$ above the recorded neurons. Source data are provided as a Source Data file.

between the changes in AC-ECoG activity and vertical depth of SD penetration, it could be further predicted that a transformation of full SDs with depression to partial SDs with no change or boom occurs gradually along the compartmentalization of SDs to superficial layers during horizontal SD propagation. This remains to be explored in future studies.

## Variety of SD patterns at the cellular level

At cellular level, SD is classically viewed as a binary process, during which neurons rapidly depolarize emitting few APs at the SD onset, and then completely lose their membrane potential and ability to generate APs due to depolarization block[5,7,65]. In the present study, neuronal behaviors during full SDs were consistent with these previous descriptions. However, membrane potential changes during partial SDs were characterized by graded levels of depolarization depending on the cell position relative to the SD stop depth, including complete depolarization in neurons located far above the SD stop depth, various levels of depolarization in neurons located close to the SD stop depth and only small change of membrane potential in neurons located far below the SD stop depth. Depolarization block also developed only in the layers invaded by SD, where the level of depolarization surpassed the AP inactivation threshold, whereas in the sub-SD zone, neurons only moderately depolarized and maintained high-frequency AP firing throughout the SD above. Therefore, neuronal behaviors during SD are variable, and neuronal depolarization can be gradual depending on the vertical profile of SD and location of the cell relative to the SD stop depth. Similar variability in the depolarization levels and excitability has also been recently described in the case of anoxic SDs in immature cortical neurons[66]. Interestingly, L5 neurons, which were primarily sampled in the present study, may extend their apical dendrites up to L1. However, their somatic membrane potential changed only slightly during partial superficial SDs presumably invading these processes. This is in keeping with previous observations showing poor propagation of depolarization from distal dendrites to soma during SD in CA1 hippocampus[27]. We suggest that mild levels of neuronal depolarization in the regions near the SD stop depth are due to a smaller increase in extracellular potassium and glutamate, a smaller decrease in extracellular calcium and less efficient ephaptic effects of small amplitude extracellular voltage shifts involved in SD generation[10,67] that also likely explains fading of vertical SD propagation during partial SDs.

## Clinical implications

Our findings have direct clinical implications for SD detection. Current algorithms for electrophysiological SD detection imply propagating depolarizing shifts during DC-ECoG recordings. In humans, too, two important exceptions to the rule that SD is always accompanied by spreading depression are already known. Thus, SDs may occur in the absence of AC activity and are then referred to in clinical jargon as isoelectric SDs characterized by a poor prognosis[68]. Because isoelectric SDs occur in electrically inactive tissue, they cannot cause spreading depression. However, as soon as they migrate into tissue where spontaneous activity still occurs, they can induce spreading depression there[12]. In addition, SDs have been described in patients in which epileptiform discharges occur superimposed particularly on the final shoulder of the negative DC shift[69]. In animal experiments, this phenomenon was originally termed "spreading convulsion" by van Harreveld and Stamm[70]. In patients, it is now referred to as "spreading depolarization with epileptiform activity" (SDEA)[71]. However, when an SD occurs in electrically active tissues, i.e., tissues with spontaneous activity, and no epileptiform activity co-occurs, it is usually not questioned by a clinician that spreading depression must be associated with it. When this is not the case and either no change in activity or a boom occurs, as explained in the present study, this fallacious belief will lead to misinterpretation. Considering these subtleties for the clinical evaluation of SDs and for the development of algorithms for automated SD detection is particularly important because SD clusters are currently the only existing real-time indicator of delayed cerebral ischemia in comatose SAH (or TBI) patients during neurocritical care and thus there may be significant consequences for the patient's outcome if SDs are missed and no treatment is initiated. Importantly, the exact location of future developing pathology is usually unknown when the neurosurgeon implants neuromonitoring devices such as a subdural electrode strip. In the recent prospective, observational, multicenter, cohort, diagnostic phase III trial, DISCHARGE-1, on SDs in 180 SAH patients, all electrodes of the subdural recording strip were located outside newly developing infarct areas in 64/90 (71.1%) patients developing delayed ischemic infarction[20]. In these patients, the newly developing infarcts are detected via SDs that have migrated out of the actual metabolically disturbed zone, and just such SDs may then also be characterized by a reduced extension in the vertical direction of the cortex, which in turn leads to no spreading depression or even a boom of spontaneous activity.

In summary, our study challenges the general concept of homology between spreading depolarization and spreading depression[2–8] by presenting evidence for a diversity of changes in cortical activity during SDs with different vertical profiles. Only full SDs strictly conform to this concept, as they are associated with an almost complete loss of membrane potential, depolarization block and suppression of activity in all layers. In contrast, partial SDs are associated with complex changes in cortical activity and the formation of three distinct vertically separated zones (Fig. 4): (i) the upper zone invaded by SD with depressed activity due to deep neuronal depolarization and AP inactivation, (ii) the intermediate sub-SD zone with increased activity caused by moderate neuronal depolarization and organized into sustained local $\gamma$-oscillations, and (iii) the deep zone that supports physiological activity and promotes, through volume conduction, electrical activity at the cortical surface, notably $\delta$–waves. Depending on the depth of SD penetration, activity in this deep zone supports different levels of electrical activity at the cortical surface, ranging from mild depression to boom. We suggest that our findings should be taken into account in the clinical SD detection protocols using recordings from the cortical surface (ECoG, EEG), and raise further questions on the role that different forms of SD play in ischemia, TBI, epilepsy and migraine aura.

## Methods
### Human study
**Patients and ECoG recordings.** A subpopulation of 31 patients from the DISCHARGE-1 study[20] was included in the analysis. This is a randomly selected third of the Berlin cohort in DISCHARGE-1 who had direct current (DC)/ alternating current (AC)-electrocorticogram (ECoG) (0–45 Hz) recordings and at least one SD[12]. Briefly, DISCHARGE-1 is a prospective, observational, multicenter, cohort, diagnostic phase III trial in which 180 patients suffering aneurysmal subarachnoid hemorrhage (SAH) were continuously monitored in the neurointensive care unit from the day of intervention up to 14 days post-hemorrhage in order to study the early and delayed development of cerebral ischemia. The main part of the monitoring consisted of continuous ECoG recording using a subdural strip of platinum/iridium electrodes (Wyler, 10 mm spacing, 5 mm diameter, Ad-Tech Medical, Racine, Wisconsin, USA) neurosurgically placed on viable cortical tissue through a craniotomy or a burr hole. A subdermal platinum/iridium needle electrode was used as a reference for monopolar recordings with a DC-coupled amplifier (BrainAmp MR plus, Brain Products, Munich, Germany) and BrainVision Recorder software (Brain Products). From this dataset of 31 aSAH patients (21 females, 10 males, mean age: $57 \pm 11$ years), all non-isoelectric SDs with good recording quality were selected in 26 patients. Around each SD, a total of 1360 fragments of 9 minutes in length were collected. The patients' ECoG were originally recorded and SDs manually detected with a Powerlab

16/SP and LabChart-8 software (ADInstruments, New South Wales, Australia) with a sampling frequency of 200 Hz. The selections of SDs from the LabChart files were exported in.txt files to later be analyzed with Python and Matlab.

## Animal study

**Surgery and animal preparation.** Wistar rats of both sexes aged from 3 to 8 weeks were used. Animals were prepared under isoflurane anesthesia at the surgical level (4% for induction, 2% for maintenance, Aerrane (Baxter, UK)), confirmed by a negative toe-pinch reflex. The skin above the skull and periosteum was removed, Hemostab (Omegadent, Moscow, Russia) was used to stop capillary bleeding and was then rinsed with 0.9% NaCl. The wound was treated with bupivacaine (0.25%). A metal ring was attached to the skull with dental cement (Meliodent, Heraeus Kulzer, Germany). Thereafter, isoflurane anesthesia was discontinued, and the animals were administered urethane (Sigma, USA) at the surgical level of anesthesia (1.5 g/kg, i.p.), confirmed by immobility, and the absence of vocalizations and negative toe-pinch reflexes during the entire experiment. During recordings, the rats were placed on a heated platform (37 °C, TC-344B; Warner Instruments, Hamden, CT). The metal ring was attached to a magnetic stand via a ball joint to restrain head movements. A chloride-coated silver wire was placed in the cerebellum to serve as a ground/reference electrode. A cranial window -0.3 mm in diameter for the placement of a silicon probe was drilled above the barrel cortex area (-2.5 mm caudal and -5.5 mm lateral from bregma). For the concomitant patch-clamp recordings, a second window -0.2 mm in diameter was made at a distance of <0.5 mm rostral from the silicone probe penetration site. A cranial window -0.3 mm in diameter for SD induction with KCl (0.2–1 M) application was made either over the occipital cortex (-6–7 mm caudal and -5-6 mm lateral from bregma) or the frontal cortex (-1–1.5 mm rostral and -3-4.5 mm lateral from bregma). The KCl application chamber was built with a 1-2 mm high dental cement wall around the cranial window[18]. Isoguvacine (10 μM, 200–600 nl, at 220 nl/min) was applied using glass pipette (tip diameter 2–3 μm) connected to the 705 Hamilton syringe (50 μl, Hamilton, USA) filled with liquid light paraffin (Panreac AppliChem, Spain) using Micro4 Microsyringe Pump Controller (World Precision Instruments, USA) at cortical depth of 1300–1400 μm at a distance of -0.5 mm from the silicone probe.

**Extracellular recordings.** Recordings of the field potential (FP) and multiple unit activity (MUA) were performed using linear multichannel silicone probes with iridium electrodes: 413 μm$^2$ surface area, 100μm separation distance (A1x16-5-100-413-A16, Neuronexus Technologies, USA). The probe was inserted vertically into the barrel cortex to a depth of 1.6–1.8 mm. The probe was coated with DiI fluorescent dye (DiI, Sigma Aldrich, USA) for the post-hoc reconstruction of the electrode position. The signals were amplified, lowpass filtered at 9 kHz and digitized at 32 kHz using a DigitalLynx SX amplifier and Cheetah 6.3.2 (Neuralynx, USA). Recordings were performed: (i) in true DC mode (input range ±131 mV) with DC potential offsets compensated at the onset of recordings[18] or (ii) using full-band recordings with inverse filtering for signal reconstruction based on hybrid AC/DC-divider RRC filters[72].

**Whole-cell recordings.** Patch-clamp recordings were performed from the barrel cortex at 1000–1300 μm depth using Axopatch 200B amplifier (Axon Instruments, Union City, CA, USA)[73]. Patch electrodes were pulled from borosilicate glass capillaries (BF150-86-10, Sutter Instrument, Novato, CA, USA) and had a resistance of 5–7MΩ. Patch pipettes were filled with a solution of the following composition (in mM): 131 potassium gluconate, 4 KCl, 10 HEPES, 10 phosphocreatine, 4 MgATP, and 0.3 Na2GTP (adjusted to pH 7.3 with KOH). The patch pipette was inserted into the cortex at -0.5 mm distance from the insertion point of the silicon probe and at a 50-65° angle towards the

silicon probe. Membrane potential values were corrected for liquid junction potential of +15 mV. Series resistance was 25–75MOhm. Patch-clamp recordings were digitized at 40 kHz with a Digidata 1440 A interface card using Clampex 10.3 (Molecular Devices, USA).

**Data analysis.** Raw data were preprocessed using a custom-developed suite of programs in the Matlab and Python environments. Positive polarity is graphed as up throughout all manuscript. In human data, SD onset was determined from the peak of the first derivative (SD′) of the 0.001-45 Hz bandpass filtered FP signal during the initial SD depolarization phase. Spectrograms of the raw DC signal in 30 s time windows were calculated in the 0.5–45 Hz range (AC-ECoG) at [-120 -90]s before the SD onset (pre-SD) and [10 40]s after the SD onset (SD) using power spectrum analysis with 5 s sliding windows and 1 s overlap. The mean power values in these thirty 5 s spectrogram fragments during the pre-SD and SD periods were then used to determine (i) the SD/pre-SD power ratio and (ii) the statistical difference in power between the pre-SD and SD periods using the Wilcoxon rank sum two-sided test. SDs with non-significant ($p > 0.05$) difference in power between the pre-SD and SD periods were considered to be SDs with "no change" in AC-ECoG activity, whereas SDs associated with a significant decrease or increase in power were considered to be SDs with a depression or boom in AC-ECoG activity, respectively. The power ratio was calculated as the mean of the power spectrum during the SD period divided by the mean of the power spectrum during the pre-SD period. The following frequency bands were used: 0.5–4 Hz for $\delta$, 8-30 Hz for $\alpha-\beta$, and 30-45 Hz for $\gamma$.

In the data obtained from rats, the original DC signal was downsampled to 1 kHz and used for the FP signal. SD onset was determined from the peak of the first derivative (SD′) of the 1 Hz lowpass filtered FP signal during the initial SD depolarization phase[18]. SD with an SD′ peak <1 mV/s were discarded from the analysis. SD stop depth was determined from the depth of the deepest electrode which displayed SD′ peak >1 mV/s. Spectral power was estimated using direct multi-taper estimators (0.5 Hz bandwidth, 3 tapers, 5 s spectral window with 1 s overlap, Chronux toolbox in Matlab) for the signal in the range of 0.5–45 Hz (AC-band) during intervals [-40 -10]s before the SD onset (pre-SD) and [10 40]s after the SD onset (SD). The SD/pre-SD ratio and the corresponding p-value were calculated as described above for the human data analysis. The UP states of $\delta-$oscillations were detected as negative events exceeding 1 STD (standard deviation) of the 0.5–4 Hz bandpass filtered FP deflections from L6. $\gamma-$troughs were detected as negative events exceeding 1 STD level from the FP signal filtered within the bandpass 30–150 Hz. Current source density (CSD) was computed according to a differential scheme for the second spatial derivative of FP along the probe axis and smoothed with a triangular kernel of length 3[74]. Spatial discrimination of FP generators of $\delta$-oscillations was performed using the kernel density Independent Component Analysis (ICA) algorithm (KDICA) that was customarily implemented in Matlab[75,76]. Recorded FP signals $u_m(t)$ were considered as the weighted sum of the activities of N neuronal sources or FP-generators:

$$u_m(t) = \sum_{n=1}^{N} V_{mn} s_n(t), \quad m = 1,2,...,M \qquad (1)$$

where ($V_{mn}$) is the mixing matrix composed of the spatial weights of N FP-generators on M electrodes and $s_n(t)$ is the time course of the $nth$ FP-generator. Using $u_m(t)$ the ICA found the ($V_{mn}$) and $s_n(t)$ and the joint group of spatial weights ($V_{mn}$) was ordered into instant depth profiles of the voltage according to electrode position. We disregarded the ICA components with a total compound variance below 1%. Consequently, in order to prioritize the FP generators for the signals of interest ($\delta$-frequency and above) the DC changes in the analyzed signal were removed prior to ICA analysis by subtracting a low-pass filtered replica of the signal through a smooth function with a time window of 0.6 times the sampling rate. When stimuli were used, short epochs

(400 ms) containing the evoked potentials were also eliminated prior to ICA not to bias the analysis of spontaneous activity by the evoked activity. Earlier we checked that maintaining the 5-6 principal components optimized the subsequent ICA separation of the more stable components along the cortical width[25]. The ICA was performed on 160 s epochs containing one SD episode. The power of each ICA-separated FP generator was evaluated in two 40 s periods selected immediately before and during SD occurrence, and skipping the high-frequency bursts heralding DC shifts. The latter periods also contained a post-DC shift fragment in which activity had not yet recovered. Note that the length of the epoch analyzed by the ICA does not modify the results noticeably.

For extracellular spike detection, the original wide-band signal was filtered (250–4000 Hz, Daubechies wavelet filter[77]), and negative local peaks >4 STD of the quietest 100 s fragment in the control (before the first SD) were considered as spikes. An adaptive threshold was then calculated for each SD based on the >4000 Hz filtered signal with a multiplier from 250 to 4000 Hz and >4000 Hz ratio for the control period to eliminate events associated with increased noise level due to the increase in electrical tissue resistance during SD[78]. MUA SD/pre-SD ratio was calculated for mean MUA frequency value within the time interval [-25 -10]s before the earliest SD onset (among all channels recording the same SD) for the pre-SD period and [0 15]s after the deepest SD onset for SD period. MUA burst duration was calculated as a total duration of MUA frequency above the mean plus 3 STD of MUA frequency in the control period. For circular statistics, the FP was filtered using a [30–150]Hz bandpass filter. Instantaneous $\gamma$−phase was computed using Hilbert transformation of the filtered signal. The phase of each detected MUA spike within a $\gamma$−cycle was derived from the instantaneous $\gamma$−phase. Phase modulation of spikes by $\gamma$−oscillations was determined using the Rayleigh circular statistics.

In whole-cell recordings, membrane potential values were corrected for the extracellular voltage shifts during SD by subtracting FP recorded by the nearby channel of the silicon probe[27]. Action potentials (APs) were detected in the 50 Hz highpass filtered signal as events exceeding the 5 mV threshold. The maximal depolarization level of neurons during SD and the threshold of AP depolarization block were calculated from a lowpass filtered (< 0.2 Hz median filter) membrane potential signal to eliminate APs. The AP burst duration during SD was calculated as the time interval between the first and last AP in the burst with a firing rate threshold of 5 APs/second.

**Statistical analysis.** Statistical analysis was performed using the Matlab Statistics toolbox. The normality assumption of the data distributions was assessed using the Shapiro–Wilk test. Since the test rejected the corresponding null hypothesis, Wilcoxon rank sum and signed-rank two-sided tests were used. For statistical comparisons between the three groups of data, the Kruskall-Wallis test was used. Wilcoxon rank sum test also was used to analyze the significance of LFP power changes between pre-SD and SD epochs. Pooled data are presented as median, 25th (Q1) and 75th (Q3) percentiles, in boxplots and violin plots. Wilcoxon rank sum test was used to assess the significance of differences between samples. Correlation coefficients were calculated as Spearman's correlation coefficient with exact p-value. The level of significance was kept at p < 0.05.

**Study approval**
The recordings from patients were carried out in accordance with protocols approved by the Charité – Universitätsmedizin Berlin, corporate member of Freie Universität Berlin, Humboldt Universität zu Berlin, and Berlin Institute of Health, Berlin, Germany (Ethical Committee Ethikausschuss CBF am Campus Benjamin Franklin, Ethical vote # EA4/022/09). Written informed consent was received prior to participation for each patient. The animal experiments were carried out in compliance with the ARRIVE guidelines. Animal care and procedures

were in accordance with EU Directive 2010/63/ EU for animal experiments, and all animal-use protocols were approved by the Local Ethical Committee of Kazan Federal University (#24/ 22.09.2020).

**Reporting summary**
Further information on research design is available in the Nature Portfolio Reporting Summary linked to this article.

## Data availability

Animal data supporting the results of the study are available from the authors upon request. Examples of the data presented in Fig. 2b are available at https://doi.org/10.5281/zenodo.10067055. Clinical data analyzed during the current study are not publicly available because the patient's informed consent only permits the data analysis and publication by the investigators. Source data are provided with this paper.

## Code availability

Codes for the data analysis are available from the authors upon request. Codes to detect SDs and analyze FP power SD/pre-SD ratios are available at https://doi.org/10.5281/zenodo.10066378.

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

## Acknowledgements

We thank Drs. G. Buzsaki, I. Timofeev, A. Sirota, R. Cossart, K. Staley and C. Bernard for their valuable comments on the manuscript, and Suchkov D., Mamleyev A. and Minlebayev M. for customizing the platform for fixation of the animal's head. This research was supported by Russian Science Foundation grant 22-15-00236 (AN, DV, RK, GBZ, KC) (electrophysiological experiments and data analysis in rodents); Deutsche Forschungsgemeinschaft grants DFG DR 323/5-1 and DFG DR 323/10-2, and Bundesministerium fuer Bildung und Forschung grant Era-Net Neuron EBio2 (JPD) (electrophysiological recordings and data analysis in patients); Ministerio de Ciencia e Innovación of Spain (MICINN) grant PID2019-111587RB-I00 (OH) (decomposition of cortical generators), and performed in the framework of the PRIORITY-2030 program of the Kazan Federal University.

## Author contributions

Conceptualization: R.K., J.P.D., O.H.; Methodology: A.N., D.V., C.L.L., G.B., K.C., J.M., O.H., J.P.D., R.K.; Investigation: A.N., D.V., C.L.L., G.B., K.C., J.M.; Visualization: A.N., D.V., C.L.L., J.M., O.H., J.P.D., R.K.; Funding acquisition: R.K., J.P.D., O.H.; Project administration: R.K., A.N., J.P.D., O.H.; Supervision: R.K., J.P.D., O.H.; Writing—original draft: R.K., A.N., J.P.D., O.H.; Writing—review & editing: A.N., D.V., C.L.L., G.B., K.C., J.M., O.H., J.P.D., R.K.. A.N., D.V. and C.L. equally contributed to the work; A.N. is listed as first co–first author because of his major contribution to the analysis; D.V. and C.L. were major contributors to the data acquisition in animal and human studies, respectively.

## Competing interests

The authors declare no competing interests.
