## [Peer Review File · Nature Communications]

Diversity of cortical activity changes beyond depression during Spreading DepolarizationsREVIEWER COMMENTS:

Reviewer #1 (Remarks to the Author):

The manuscript titled "Diversity of cortical activity changes beyond depression during spreading depolarizations" by Nasretidinov et al reports SDs inducing varying responses in electrical activity on cortical surface recordings (i.e., depression, no changes, and increases). The majority of recordings show typical depression after SD in the cortical surface recordings, however, a few show no changes or increases in activity (booming). In these recordings, the activity in the delta frequency band (0.5-4Hz) is enhanced, particularly during SDs that do not travel throughout the entire cortex (i.e., partial SDs). The authors first show these variable responses during SD depolarization phase (DC shift) in subarachnoid hemorrhage patients. They then observe similar phenotypes in naïve animals with SD evoked with KCl. The authors suggest that this variability is inherently associated with SD and not dependent on disease models.

The authors use multiple unit activity (MUA) recordings to show increases in gamma oscillations in deeper layers (layers 5/6) when SD propagation stops at the superficial layers. However, it is not clear whether these increases in action potential firing in deeper layers correspond to increases in activity seen in 'up states' from surface recordings. It would be useful to show 0.5-4 Hz LFP activity along with MUA in Figure 4 (same as Figure 1).

Further, the authors use current source density and independent component analysis on the field potential profile of the 'up states' and find that currents generated in deeper layers may serve as the source for the surface activity via volume conduction of signals. Additional experiments could substantiate the direct involvement of deeper layers in the activity in surface cortical recordings during partial SDs. For example, hyperpolarizing layer 5 neurons directly or with archaerhodopsin and halorhodopsin could provide some mechanistic evidence and increase the explanatory potential of the authors' findings. As it stands, these are interesting, well analyzed phenotypes, but the work is descriptive.

Specific notes to consider:

The term 'booming' may be unfamiliar – perhaps just describe an increase in activity? Or define the term earlier in the text.

Fig 4 shows different phenotypes of membrane depolarization during SD in layer 5 neurons but the baseline activity is different between comparisons (see Fig. 4C). We know that there are regular spiking and intrinsic bursting pyramidal cells in layer 5, and they have differential physiological and anatomical properties. It is important to show baseline recordings of at least 3 min and neuronal membrane properties for all phenotypes in Fig 4C.

The authors state "SDs stopping far above the recorded neuron were associated with little or no change in membrane potential and neuronal firing (Fig. 4C, SD#5 & Fig. 6)". However, Figure 6E shows small but significant depolarization (compared to pre-SD) in layer 5 neurons when SDs do not propagate throughout the cortex. Please discuss.

Traces from the surface do not show increases in delta power or up states in Figure 5C. It would be informative to show MUA and LFP (including 0.5-4Hz) activity for all cortical layers in cases of partial SDs associated with a transient increase in cortical activity.

Also add layer 5 neuronal membrane properties including membrane resistance, resting membrane potentials, etc. Changes in these properties during partial SDs may affect up states and gamma oscillations.

Add statistical methods in the results or in the figure legend. On a similar note, the authors mention using Wilcoxon rank sum test to assess the significance of differences between samples, however there are comparisons between three groups are performed (e.g., Figure 3F). This needs to be addressed.

Need to discuss power analyses and normality tests.

Instead of maximal depolarization noted in the results, suggest reporting changes in depolarization during full/partial SDs.

Instead of control, use "pre-SD" as this is what is being compared.

Reviewer #2 (Remarks to the Author):

The authors show that Spreading Depolarizations (SDs) in humans with subarachnoid hemorrhage display a variety of electrophysiological activity beyond the commonly associated spreading depression. They then show that these findings can be reproduced in a rodent model, allowing for greater recording access and experimental control to probe how the variety of electrophysiological signals relate to features of the SDs. Importantly, they find that partial SDs (i.e. those which fail to penetrate the entire depth of cortex) result in activity which deviates from the traditional spreading depression due to sparing of delta activity located deeper than the SD was able to penetrate. They further show that propagation/penetration of both partial and full SDs are mediated by gamma waves, and in the partial SD case, this culminates in a sustained gamma oscillation in the sub-SD region. Finally, they demonstrated that the membrane potentials of individual L5 cells differ in the partial vs full SD case, with partial SDs resulting in a slight, sustained depolarization that can contribute to the sustained gamma activity and occasional booms the authors observed during partial SDs.

Overall Impression:

The authors do a very nice job at addressing this important issue which impacts both how research into SDs is done at a more basic science level, as well as astute recommendations for translational research and clinical practice. Moreover, the correspondence between the clinical population data and the rodent model data is impressive. The paper is written clearly (with one exception - see minor critique 1), and the authors do a good job of pointing to the clinical and translational importance/application of this work without overselling it. The paper brings to light an important issue in clinical neuroscience and neurology, and acts as a demonstration for phenomenal translational research.

Critiques:

1. In the Discussion, beginning on page 8, it is stated that "during partial SDs, only superficial δ -generators were selectively suppressed whereas deep δ -generators persisted". Considering that cortical slow oscillation (δ -waves) are presumably led by deep layer 5 pyramidal cells, preserving the deep layers activity would definitely explain persistence of δ -oscillations, consistent with study results. It does not explain, however, why activity may increase ("boom") in some cases of SD and this increase was specifically reflected in δ -oscillation band. Does it mean that slow oscillation increase in deep layers over control in such cases of SD?

2. Some studies suggested that cortical slow oscillation may depend significantly on thalamocortical projections that help to synchronize cortical slow waves. Furthermore, other works propose contribution from thalamic mechanisms of δ -activity which depend on a few calcium and voltage dependent ion channels (e.g. the IT and I_h currents) in thalamic neurons. Given these ideas about role of thalamus (direct or indirect) in generating cortical slow oscillation, is it possible that the Remote generator described in the study is related to thalamic-generated delta? That would possibly suggest why it is spared during SDs - it is both anatomically distinct (as the Remote generator likely is), but it is also generated through a distinct mechanism that involves thalamocortical projections. Some discussion of these possibilities would be appropriate.

Minor Critiques:

1. The section Neuronal activity during SDs with different vertical propagation profiles is somewhat difficult to follow. The current order in which results are discussed requires jumping back-and-forth between Figs 4, 5, & 6. If it's possible to refer to Figures (at least the first time a subpanel is introduced) in a more linear order, it would make understanding the main messages of each figure much easier.

2. Page 11 lines 15-16 missing an open parenthesis

RESPONSE TO REVIEWERS

Ms# NCOMMS-23-12488A

Diversity of cortical activity changes beyond depression during spreading depolarizations

REVIEWER COMMENTS

Reviewer #1 (Remarks to the Author):

The manuscript titled “Diversity of cortical activity changes beyond depression during spreading depolarizations” by Nasretidinov et al reports SDs inducing varying responses in electrical activity on cortical surface recordings (i.e., depression, no changes, and increases). The majority of recordings show typical depression after SD in the cortical surface recordings, however, a few show no changes or increases in activity (booming). In these recordings, the activity in the delta frequency band (0.5-4Hz) is enhanced, particularly during SDs that do not travel throughout the entire cortex (i.e., partial SDs). The authors first show these variable responses during SD depolarization phase (DC shift) in subarachnoid hemorrhage patients. They then observe similar phenotypes in naïve animals with SD evoked with KCl. The authors suggest that this variability is inherently associated with SD and not dependent on disease models.

RESPONSE:

We would like to thank the reviewer for carefully evaluating our work and valuable suggestions and comments.

Comment #1

The authors use multiple unit activity (MUA) recordings to show increases in gamma oscillations in deeper layers (layers 5/6) when SD propagation stops at the superficial layers. However, it is not clear whether these increases in action potential firing in deeper layers correspond to increases in activity seen in 'up states' from surface recordings. It would be useful to show 0.5-4 Hz LFP activity along with MUA in Figure 4 (same as Figure 1).

RESPONSE:

According to this advice (and also in response to the comment #6 below), in addition to the 1.5 s-long full-band LFP fragments #1 and #2 from the pre-SD and SD epochs on Fig. 4a, in the Figure 4, we have also:

- 1) added new panel **c**, which shows 0.5-4 Hz LFP and quantifies delta and gamma power and MUA through the entire time course of partial SD shown on panel **a** at three depths: (i) at the top electrode (in the MUA depression zone), (ii) at the depth of 1100 microns (in the Sub-SD excitation zone), and (iii) at the depth of 1700 microns (zone of unchanged MUA);
- 2) added new panel **d**, which provides SD/pre-SD ratio for delta and gamma power at different cortical depths relatively to the SD stop depth;
- 3) added panel **b** to illustrate local nature of gamma oscillations in the Sub-SD excitation zone formed during partial SDs.

In the discussion, we also added the following sentence (Pg. 10 Ln 277):

“Of note, because of the limited extent of γ -oscillations, their volume-conducted signals poorly propagate at a distance and are therefore barely visible at the cortical surface^{25,60}.”

Altogether, these results show that excitation in the Sub-SD zone, organized in local gamma oscillations, does not correspond to increase in delta-activity and UP-states during partial SDs at the surface.

Comment #2

Further, the authors use current source density and independent component analysis on the field potential profile of the ‘up states’ and find that currents generated in deeper layers may serve as the source for the surface activity via volume conduction of signals. Additional experiments could substantiate the direct involvement of deeper layers in the activity in surface cortical recordings during partial SDs. For example, hyperpolarizing layer 5 neurons directly or with archaerhodopsin and halorhodopsin could provide some mechanistic evidence and increase the explanatory potential of the authors’ findings. As it stands, these are interesting, well analyzed phenotypes, but the work is descriptive.

RESPONSE:

We fully agree with the reviewer, and performed additional experiments with silencing deep layers using microinjection of isoguvacine at cortical depth. These new results are presented in the new Supplementary Figure 2 and related text as follows (Pg. 6, Ln.143):

“To verify this hypothesis, we inhibited deep cortical layers by local injection of the GABA(A) receptor agonist isoguvacine. Isoguvacine microinjection at depth ~ 1.3 mm induced long-lasting MUA suppression in deep cortical layers (to $<10\%$ of control values) without significant change in MUA levels in the superficial layers (Supplementary Fig. 2a,b), and suppressed δ -FP power and δ -generators through all layers (Supplementary Fig. 2c,d). Isoguvacine also strongly reduced the changes in δ -activity both during full and superficial SDs, and eliminated an increase in δ -activity at the top electrodes during superficial SDs (Supplementary Fig. 2e,f) supporting the hypothesis that persisting δ -activity at the cortical surface during partial SDs in control conditions is supported by spared activity in deep layers.”

Specific notes to consider:

Comment #3

The term ‘booming’ may be unfamiliar – perhaps just describe an increase in activity? Or define the term earlier in the text.

RESPONSE:

According to this advice, we introduce the term “boom” in the main text as follows (Pg. 3, Ln. 64):

“...a paradoxical transient increase in cortical activity, hereafter referred to as a “boom”

Comment #4

Fig 4 shows different phenotypes of membrane depolarization during SD in layer 5 neurons but the baseline activity is different between comparisons (see Fig. 4C). We know that there are regular spiking and intrinsic bursting pyramidal cells in layer 5, and they have differential physiological and anatomical

properties. It is important to show baseline recordings of at least 3 min and neuronal membrane properties for all phenotypes in Fig 4C.

RESPONSE:

In keeping with this advice, we have made the following changes in the manuscript:

- 1) In the Figure 5 we specify that all four SDs were recorded from the same L5 neuron, which has been identified as intrinsic bursting cell (cell #3 in the Supplementary Table 1). Four SD examples presented in this figure (a-d) were selected from a cluster of 14 SDs, and each SD # now corresponds to its # in the cluster indicated in the Supplementary Table 2. In the figure 5, these four SDs are ranged along with SD stop depth. We also expanded time of the total trace. Cell identity (L5 IB neuron #5) is also indicated in the Figure 6.
- 2) Neuronal membrane properties for all neurons are now presented in Supplementary Table 1. On the basis of firing properties and action potential waveforms, 5 cells were identified as regular spiking cells, 4 cells as intrinsic bursters, and one cell as a fast-spiking cell (this information is added to the text, Pg. 7 Ln. 165).
- 3) The new supplementary Table 2 describes the behavior of cells during all SDs of different stop depth, including their depolarization levels and AP firing.

Comment #5

The authors state “SDs stopping far above the recorded neuron were associated with little or no change in membrane potential and neuronal firing (Fig. 4C, SD#5 & Fig. 6)”. However, Figure 6E shows small but significant depolarization (compared to pre-SD) in layer 5 neurons when SDs do not propagate throughout the cortex. Please discuss.

RESPONSE:

We fully agree with the reviewer, and this statement has been modified to as follows (Pg. 7 Ln. 177):

“SDs stopping far above the recorded neuron were associated with little depolarization through the SD above and small reduction in the UP-states amplitude (Fig. 5d,e,f & Fig.6)”

Comment #6

Traces from the surface do not show increases in delta power or up states in Figure 5C. It would be informative to show MUA and LFP (including 0.5-4Hz) activity for all cortical layers in cases of partial SDs associated with a transient increase in cortical activity.

RESPONSE:

MUA and LFP (including 0.5-4Hz) are now shown at three depths on panel Fig. 4c (together with delta and gamma power), and group data on delta and gamma power are presented on Fig. 4d with 100 microns spacing relatively SD stop depth. Please also see our response to the comment #1.

Comment #7

Also add layer 5 neuronal membrane properties including membrane resistance, resting membrane potentials, etc. Changes in these properties during partial SDs may affect up states and gamma oscillations.

RESPONSE:

Neuronal membrane properties are summarized in the Supplementary Table 1. Additional Supplementary Table 2 also describes behavior of these cells during all SDs of different stop depth including their depolarization levels and AP firing. Also, cell types are indicated in the Figures 5 and 6, and the text has been updated accordingly. Please see also response to comment #4 above.

Comment #8

Add statistical methods in the results or in the figure legend. On a similar note, the authors mention using Wilcoxon rank sum test to assess the significance of differences between samples, however there are comparisons between three groups are performed (e.g., Figure 3F). This needs to be addressed. Need to discuss power analyses and normality tests.

RESPONSE:

We have added information about the statistical methods used in the descriptions of the main and supplementary figures. For statistical comparisons between three groups of data we have used Kruskal-Wallis test. Updated statistical comparison results are added in Figures 2f and 3f, and corresponding descriptions in the Figures Legends and Materials and Methods sections. The normality assumption of the data distributions was assessed using the Shapiro–Wilk test. Since the test rejected the corresponding null hypothesis, Wilcoxon rank sum and signed rank tests were used. Wilcoxon rank sum test also was used to analyze the significance of LFP power changes between pre-SD and SD epochs. This information has been added to the figure legends and is also presented in the description of the power ratio calculation in the Materials and methods data analysis part.

Comment #9

Instead of maximal depolarization noted in the results, suggest reporting changes in depolarization during full/partial SDs.

RESPONSE:

We have modified the manuscript in line with this advice. In addition to absolute E_m values, we also show depolarization relative to pre-SD E_m values in the figures 5f,g and 6f and related text as follows:

(Pg. 7 Ln. 170):

“Full SDs and SDs that propagated into the cortex deeper than the site of the cell being recorded were associated with profound depolarization to -26 ± 15 mV (by 34 ± 13 mV from the pre-SD values, $n=26$ SDs).”

(Pg. 7 Ln. 177):

“However, during partial SDs stopping in their vertical propagation just above the neuron being recorded, membrane potential exhibited only moderate depolarization and neurons displayed sustained AP firing throughout the duration of the SD above (Fig. 5c & Fig. 5g) with maximal depolarization to -47 ± 6 mV (by 18 ± 4 mV from the pre-SD values, $n=11$ SDs; Fig. 5e).”

Comment #10

Instead of control, use “pre-SD” as this is what is being compared.

RESPONSE:

We have made the modifications requested in the figures and text of the manuscript.

Reviewer #2 (Remarks to the Author):

The authors show that Spreading Depolarizations (SDs) in humans with subarachnoid hemorrhage display a variety of electrophysiological activity beyond the commonly associated spreading depression. They then show that these findings can be reproduced in a rodent model, allowing for greater recording access and experimental control to probe how the variety of electrophysiological signals relate to features of the SDs. Importantly, they find that partial SDs (i.e. those which fail to penetrate the entire depth of cortex) result in activity which deviates from the traditional spreading depression due to sparing of delta activity located deeper than the SD was able to penetrate. They further show that propagation/penetration of both partial and full SDs are mediated by gamma waves, and in the partial SD case, this culminates in a sustained gamma oscillation in the sub-SD region. Finally, they demonstrated that the membrane potentials of individual L5 cells differ in the partial vs full SD case, with partial SDs resulting in a slight, sustained depolarization that can contribute to the sustained gamma activity and occasional booms the authors observed during partial SDs.

Overall Impression:

The authors do a very nice job at addressing this important issue which impacts both how research into SDs is done at a more basic science level, as well as astute recommendations for translational research and clinical practice. Moreover, the correspondence between the clinical population data and the rodent model data is impressive. The paper is written clearly (with one exception - see minor critique 1), and the authors do a good job of pointing to the clinical and translational importance/application of this work without overselling it. The paper brings to light an important issue in clinical neuroscience and neurology, and acts as a demonstration for phenomenal translational research.

We are grateful to the reviewer for the appreciation of our work.

Critiques:

Comment #1

1. In the Discussion, beginning on page 8, it is stated that “during partial SDs, only superficial δ -generators were selectively suppressed whereas deep δ -generators persisted”. Considering that cortical slow oscillation (δ -waves) are presumably led by deep layer 5 pyramidal cells, preserving the deep layers activity would definitely explain persistence of δ -oscillations, consistent with study results. It does not explain, however, why activity may increase (“boom”) in some cases of SD and this increase was specifically reflected in δ -oscillation band. Does it mean that slow oscillation increase in deep layers over control in such cases of SD?

RESPONSE:

We added discussion of this issue in the revised manuscript as follows (Pg. 9 Ln. 250):

“A paradoxical boom in δ -band activity at the cortical surface, which was characteristic of partial SDs with minimal SD penetration, remains only partially understood, however. The parsimonious explanation could involve the suppression of superficial, electronegative at cortical surface dipoles during partial SDs, which partially mask surface-positive deep UP-state dipoles^{25,54}. Furthermore, superficial SDs have been associated with an enhancement of deep L5/6 δ -generator. However, FP δ -power and neuronal depolarization during UP-states in deep neurons slightly decreased during superficial SDs. Therefore, an apparent increase in L5/6 cortical δ -generator could be due to suppression of the superficial component of the Main generator and leakage of its deep component to the L5/6 generator. Alternatively, suppression during partial SD of complex multipolar currents associated with the Main generator would unmask the actual (larger) magnitude of currents associated with the spared L5/6 cortical δ -generator, supporting both the apparent increase of this generator and the paradoxical boom of δ -activity during superficial SDs^{25,51}. In future research, it would be interesting to verify the mechanisms underlying SD stop depth-specific changes in cortical activity across cortical layers, including the activity boom at the cortical surface during partial SDs using a computational modeling approach^{55,56 10,25,57,58}.”

Comment #2

2. Some studies suggested that cortical slow oscillation may depend significantly on thalamocortical projections that help to synchronize cortical slow waves. Furthermore, other works propose contribution from thalamic mechanisms of δ -activity which depend on a few calcium and voltage dependent ion channels (e.g. the IT and Ih currents) in thalamic neurons. Given these ideas about role of thalamus (direct or indirect) in generating cortical slow oscillation, is it possible that the Remote generator described in the study is related to thalamic-generated delta? That would possibly suggest why it is spared during SDs - it is both anatomically distinct (as the Remote generator likely is), but it is also generated through a distinct mechanism that involves thalamocortical projections. Some discussion of these possibilities would be appropriate.

RESPONSE:

We fully agree to the thalamocortical mechanisms as explained by the reviewer. In the revised text we have made the following changes according this comment:

1) Pg. 9 Ln. 225:

“ δ -oscillations are internally generated in cortical and thalamocortical networks and support large-scale horizontal synchronization of cortical activity^{31-34 35-38}. They are supported by collective fluctuations of cortical neurons between UP and DOWN states driven by intracortical excitatory and inhibitory synaptic connections and thalamocortical inputs^{34,38-45}.”

2) Pg. 9 Ln. 236:

“Spared during partial SDs, the deep δ -generator likely relies on minimally altered activity in the deep sub-SD zone, supported by spared local circuitry and thalamocortical inputs.”

Concerning the origin of the remote generator, it is unlikely to arise from the thalamus, at least in rodents. The cytoarchitectonic features are unfavorable (multipolar neurons and absence of ordered cell aggregates). In our former explorations the bulk of thalamic FP activity was found to be volume

conducted, either from cortical sites (e.g. delta activity) or from the dentate gyrus (e.g., alpha or theta oscillations (Torres et al., 2019; Ref 25). We found a very tiny spatially-broad FP generator in the thalamus (unpublished) of an amplitude even smaller than the cortical remote generator. In primates, and cats, the cytoarchitecture is much more favorable, particularly in the lateral geniculate nucleus. In awoken monkey (*Macaca mulatta*) we found very nice local generators in this thalamic nucleus (Makarova et al., 2014; doi: 10.3389/fnsys.2014.00066; PMID: 24822038). Therefore, we cannot completely rule out this possibility in humans that present even more favorable cytoarchitecture. Yet, in the present case in rodents, the remote generator more likely arises from the hippocampus, even some theta could be occasionally appreciated, as explored in detail (Torres et al., 2019; Ref 25).

Comment #3

Minor Critiques:

1. The section Neuronal activity during SDs with different vertical propagation profiles is somewhat difficult to follow. The current order in which results are discussed requires jumping back-and-forth between Figs 4, 5, & 6. If it's possible to refer to Figures (at least the first time a subpanel is introduced) in a more linear order, it would make understanding the main messages of each figure much easier.

RESPONSE:

Following this advice, we have reorganized the figures:

- 1) The most essential data from previous Figure 5 describing gamma oscillations at the SD front (pre-SD excitation) and in the sub-SD excitation zone have been moved to Figure 4.
- 2) Figures 5 and 6 describing whole-cell data now appear sequentially at the end of the results section.

Comment #4

2. Page 11 lines 15-16 missing an open parenthesis

RESPONSE:

Corrected

REVIEWERS' COMMENTS

Reviewer #2 (Remarks to the Author):

The revised version addressed my concerns.